# Analysing Monetary Policy Shocks by Sign and Parametric Restrictions: The Evidence from Russia

**Bünyamin Fuat Yıldız [1], Korhan K. Gökmenoğlu [2] and Wing-Keung Wong [3,4,5,*]**

[1] Departamento de Fundamentos del Análisis Económico (FAE), Universidad de Alicante, 03690 Alicante, Spain
[2] Department of Banking and Finance, Eastern Mediterranean University, Famagusta 99628, Turkey
[3] Department of Finance, Fintech & Blockchain Research Center, and Big Data Research Center, Asia University, Taichung City 41354, Taiwan
[4] Department of Medical Research, China Medical University Hospital, Taichung City 40402, Taiwan
[5] Department of Economics and Finance, The Hang Seng University of Hong Kong, Hong Kong 999077, China
* Correspondence: wong@asia.edu.tw

**Abstract:** Most, if not all, of the studies in the existing literature that have examined the impacts of monetary policy implications on macroeconomic aggregates suffered from misleading impulse responses. To overcome the limitations in the existing literature and to fill the gap in the literature, this study applies the new Keynesian model by imposing the sign and parametric restrictions to investigate the effects of policy shocks on the economic aggregates for Russia by implementing SVARs, yielding a better understanding of the impacts of monetary policy shocks on the Russian economy and proving superior to other existing methods. Our approach avoids impulse response anomalies such as the price puzzle and eludes implausible overshooting responses to the subjected innovations by using prior information. Our findings indicate that although monetary policy shocks create a significant decrease in inflation in the short run within both median target responses and median responses, they have a tolerable negative effect on the output gap. On the other hand, demand shocks do not generate a significant rise in output but create inflation, while cost–push shocks generate significantly detrimental results in both inflation and output. The results draw a further step towards validating the new Keynesian theory in the Russian case by revealing the short-run nonneutrality of monetary policy intervention. Our findings also showed that the cost–push shocks have significant damaging effects on both inflation and output and that interest rates strongly respond to both cost–push and demand shocks. Our findings successfully solve the price puzzle problem, justify the new Keynesian theory that holds that monetary policy shocks only have a short-run effect, and imply that Volcker–Greenspan's rule could be a useful guide for policy makers to solve the problem efficiently. In addition, our findings can be used to make important policy recommendations for policy makers as discussed in the conclusion section.

**Keywords:** monetary policy; new Keynesian model; sign-restricted SVARs

**JEL Classification:** C51; E12; E52



## 1. Introduction

The influence of structural shocks on economic variables must be parsimoniously monitored and adequately managed to support a sustained economy. However, in analyzing the impact of monetary policy shocks on inflation, there might be misleading visualizations of impulse responses, which are defined as the "price puzzle" problem: that is, monetary policy shocks raise inflation. To overcome this predicament, economists have proposed several methodological modifications, such as reflecting theoretical knowledge as a parametric constraint on shocks or applying the restrictions solely based on signs of responses. Recently, a new perspective was introduced by Ouliaris and Pagan (2016) that brings together sign and parametric restrictions. This approach allows for obtaining

theoretically compatible impulse responses that are freed from the price puzzle problem. Additionally, the mentioned approach provides a more comprehensive depiction of the mechanism by which monetary policy shocks spread over other macroeconomic aggregates, thereby enabling us to better understand the dynamics of subjected shocks. In this study, we investigated the effects of monetary policy shocks on the Russian economy in the context of a new Keynesian macroeconomic model using the recent approach of Ouliaris and Pagan (2016). Although Russia has inflicted numerous shocks since the late 1990s and is an interesting case for monetary policy research, it has been widely neglected by researchers. Our study aims to contribute to the literature by better observing the interactions of structural shocks and providing more reliable policy recommendations.

One of the primary assumptions of monetary policy emphasizes that an unexpected increase in the short-term interest rate will reduce inflation. An increase in prices following a rise in interest rates causes a phenomenon called a price puzzle. Since the pioneering studies of Christiano and Eichenbaum (1992) and Sims (1992), the unexpected response of price level to monetary policy shocks has attracted the attention of many researchers, who have put forward several explanations. The omission of the central banks' prospects in VARs is one of the most widely offered explanations (Gan and Soon 2003; Castelnuovo 2012). According to this view, if a central bank expects an increase in inflation, it tends to raise the interest rate as a preventive measure. However, a measure taken by the central bank has a lagged effect on the targeted variable; that is, inflation will persist for some more time following the interest rate increase, and both variables have high levels for this period. Hence, a rise in the interest rate will create the impression that this measure causes an increase in inflation. In this respect, the misunderstanding created by this synchronicity appears to be the main cause of the problem, and the price puzzle is created by the noninclusion of the central banks' prospects in the VAR.

To solve the price puzzle, researchers proposed several recommendations (for early seminal papers, see Sims 1992; Balke and Emery 1994; Grilli and Roubini 1995; Cecchetti 1996; Leeper 1997). Sims (1992) offered the inclusion of commodity prices in a model to avoid inconsistent results in VARs, thereby solving the price puzzle. Following Sims's recommendation, Rusnák et al. (2013) included commodity prices in their model but failed to avoid contradictory price puzzle results for half of their estimates. Another proposal aimed to solve the price puzzle based on putting restrictions on the signs of the variables, which was introduced in some of the earliest studies in this area, including by Faust (1998) and Uhlig (2005). In addition to these approaches, Canova and De Nicolo (2002), Cho and Moreno (2002), and Sims and Zha (2006) offered another approach that uses parametric restrictions to obtain more plausible impulse response functions. However, all the mentioned approaches contain inefficiencies that we will refer to in detail in the literature review section. Recently, Ouliaris and Pagan (2016) developed a new approach that can solve the price puzzle problem and present better observations of the policy effects than preceding applications.

Russia is among the biggest economies in the world and one of the dominant forces in global economic politics. Russia experienced a transformation period following the disintegration of the Soviet Union in 1991 and applied many economic reforms later on. In 2007, the Bank of Russia declared inflation targeting as its primary policy goal. For several years following the announcement, Russia followed reassuring macroeconomic management strategies that consolidated the strict monetary stance with fiscal austerity to diminish uncertainty and mitigate the impacts of adverse external shocks. However, after 2014, increasing geopolitical tensions exposed Russia to a series of economic sanctions creating an adverse effect on the economy. In addition, global factors driving a drop in oil prices increased the uncertainty of the Russian economy and resulted in a new restraint. In 2017, necessary institutional measures were taken as a response to the mentioned external shocks and increased the resistance of the Russian economy at least in part. These institutional measures ensured confidence through the transparency of the Bank of Russia and its devotion to a tight monetary stance, which aimed to reduce inflation. The peculiar

conditions arising due to the mentioned historical developments had consequences for the administration of the monetary policy of Russia and made the country an interesting case for monetary policy research.

Following the recent turbulence experienced by Russia, several researchers investigated the interaction of structural shocks, their effects on the economy, and subsequent monetary policy responses. Mironov (2015) mentioned that the Bank of Russia (CBR) implemented a contractionary monetary policy in 2015 following a cost–pushing shock. Although this precautionary intervention decreased inflation to a certain degree, household spending diminished critically, causing a decline in national output. Fal'tsman (2016) stated that lead investments decreased by 10% due to a drop in lending following a sudden increase in the Central Bank's monetary policy rate, which created discussions concerning the efficiency of monetary policy in Russia. Ilyashenko and Kuklina (2017) reported that the move of the CBR to lower the inflation rate created a sharp drop in national output. Ivanova (2016) provided evidence for the presence of cost–push shocks in inflation formation in Russia. Bhattarai (2016) stated that adverse supply-side shocks surpass the power of policy makers; hence, monetary policy measures create an effect on the aggregate demand only in the short term. Despite achieving steadiness in the price level, cost–push shocks create challenging policy trade-offs for policy-makers' sacrifices from growth. Thus far within the Russian literature, the role and efficiency of monetary policy have not been investigated with contemporary approaches. Hence, we aimed to clarify the role of monetary policy on the Russian economy by employing a recently developed method that is based on sign and parameter restrictions (Ouliaris and Pagan 2016).

In this context, this study presents several noteworthy contributions to the literature. Primarily, the contemporary approach of Ouliaris and Pagan (2016) proposes integrated modelling of sign restriction with parameter restrictions. This approach has many desirable properties and possesses some benefits over earlier approaches in several respects. First, it is more efficient when there are long-run and zero restrictions on instantaneous responses, including further lags. During the process of generating impulse responses captured by structural estimation, the results of structural responses are appraised by the sign restrictions. Second, the theoretical foundations of the parametric constraints are based on Cho and Moreno (2002), who give a new Keynesian economic theory background based on rational expectations that allow us to make inferences concerning the theory. Thus, it is possible to examine the effects of monetary policy applications on a temporal basis in the cyclical fluctuations in output gap and inflation. Third, this approach allows researchers to elude implausible overshooting responses, avoid impulse response anomalies, and obtain theoretically compatible responses. Hence, it offers a solution to the price puzzle. In addition, this approach provides a more comprehensive understanding of the interactions among the shocks and their spread and dynamics. Then, it is possible to obtain more plausible empirical findings and in turn provide more effective policy recommendations. Another contribution of our study is that we examine an important but widely neglected case. As the beforementioned studies emphasize, there is a need for analyzing the transmission mechanism of monetary policy shocks in the rebalancing process of the output gap and inflation in Russia. Accordingly, this study enables us to examine whether activist policy making corrects the market efficiently. It also allows us to monitor the adjustment speed of demand and cost–push shocks facing Russia. As a result, we find that the actual effects of nominal shocks depend largely on the rate of adjustment speed of prices. Consequently, the information that emerged from this investigation aimed to provide more reliable evidence regarding the rebalancing process including useful implications for policy making.

## 2. Literature Review

The first part of this section discusses the development of restriction strategies conducted in VARs to reflect theoretical knowledge and thereby avoid the price puzzle. Those strategies are mainly divided into two groups, namely, parametric restrictions and sign restrictions, and they provide better results than preceding monetary policy research that

employed standard VARs. The last part of this section highlights the studies that analyze several aspects of monetary policy shocks in the case of Russia.

To prevent misconceptions about monetary policy shocks, such as the price puzzle, and obtain theoretically more plausible empirical findings, researchers first offered to use restrictions based solely on signs of the variables of the VARs. Faust (1998) introduced the concept of sign restrictions with two different VAR models to investigate the impacts of monetary policy shocks on the main economic aggregates. He found out that monetary policy does not create any significant impact on the main economic aggregates on output. Canova and De Nicolo (2002) examined the impacts of monetary policy shocks and concluded that these shocks affect inflation and output cycles significantly in all G7 countries. Uhlig (2005) found that monetary policy has no significant impact on real gross domestic product fluctuation but does affect price levels in the US. He rejected the short-run neutrality of monetary policy implications. Scholl and Uhlig (2008) analyzed the propagation mechanism of monetary policy shocks and found that they resulted in an appreciation in exchange rates. In the same vein, Ho and Yeh (2010) imposed sign restrictions for Taiwan using six fundamental economic variables and reported that contractionary policies affect output persistently. The persistent negative impact of monetary policy shocks is also recognized in inflation. Analysing four developed countries, Kim and Lim (2018) concluded that monetary policy shocks lead to significant exchange rate appreciation in all subjected countries.

Parameter restriction is the other approach offered to contribute to solving the misconception aroused by basic VAR models (for some early examples, see Cushman and Zha 1997; Bernanke and Mihov 1998). In one well-known early study investigating the literature on monetary policy implications, Christiano et al. (1999) reported that there is no agreement concerning the effects of monetary policy shocks. Christiano et al. (1999) suggested that if the researcher makes enough restrictions to isolate other shocks affecting the economy, it is possible to observe systematic changes caused by monetary policy shocks. In this vein, Cho and Moreno (2002) introduced a restricted parameter model based on a new Keynesian framework within the concept of rational expectations. They developed their research further and concluded that monetary policy interventions have no significant impacts on output (Cho and Moreno 2006). Sims and Zha (2006) employed an SVAR model to better understand whether monetary policy can reduce inflation or output fluctuations without any essential cost and concluded that the real impact of monetary policy is limited. Berument (2007) used parametric restrictions to trace the impacts of monetary policy shocks for Turkey. He reported that monetary policy shock has a temporary effect on output, but the decrease in inflation seems persistent.

Although they are limited in number, there are also researchers who employed both parameter and sign restrictions to analyse the impacts of monetary policy shocks on macroeconomic aggregates. The innovative works of Baumeister and Benati (2012) pioneered a multivariate approach using combinations of both sign and parametric restrictions. They found out the long-term bond yield has a significant impact on output and inflation for the U.S., Europe, Japan, and the United Kingdom. Bjørnland and Halvorsen (2014) investigated the relationships between monetary policy shocks and exchange rates in six developed countries. Their findings pinpoint the similarities between four of the six countries' monetary policy responses to exchange rate shocks. Haberis and Sokol (2014) proposed a new model for analysinge the interactions between technology, government spending, and monetary policy shocks using sign and parametric restrictions.

Carrillo and Elizondo (2015) analysed the efficacy of parametric and sign restrictions to estimate the impacts of monetary policy shocks in Mexico. Their study showed that using solely sign restrictions regularly overshoots inflation responses to the monetary policy shock. In addition to this, the authors stated that similar shortcomings can occur with only parametric restrictions; that is, this approach also generates high responsiveness of the impulse responses to monetary policy shocks. In this direction, they proposed attaching prior knowledge to narrow the amount of economically unreasonable responses.

As the authors state, their aforementioned modification robustly improves the propagation mechanism of the shocks. Utilizing the same method, Njindan Iyke (2016) found that after a monetary policy shock, interest rates exhibited a significant rise, but the nominal exchange rate showed appreciation in two forecast horizons for Nigeria. Benati (2015) examined the interaction of unemployment with monetary policy shocks in five countries in Europe and found that the impacts of monetary policy shocks on unemployment were negative and displayed very similar economic features within each sampled country. The authors also acknowledged that inflationary shocks do not trigger a permanent reduction in the unemployment rates of any subjected economy.

Ouliaris and Pagan (2016) provided a novel approach that predicts structural equations via instrument variables. To commence the signing procedure, it follows the given structural restriction imposed. The computational requirements of this process are related to the method introduced by Fry and Pagan (2011). However, while Fry and Pagan (2011) based theirs on the utilization of rotation matrices, Ouliaris and Pagan (2016) utilized any simultaneous equation arrangements and included a broader spectrum of information on both the parameters and IRs, which made their approach more credible. Following this methodological proposition, they attained more plausible results by repeating the study of Cho and Moreno (2006), who had built on assumptions that innovations of the monetary policy and cost–push shock equations zero long-run effects on the output gap. In the same vein, Fisher and Huh (2016) found that in the median responses, there were no perplexing results for exchange rates. Concerning Europe, following a monetary policy shock, there was limited appreciation. Fisher and Huh (2019) performed Ouliaris and Pagan's (2016) model considering oil price shocks within the structural framework developed by Peersman (2005). They successfully avoided the price puzzle problem and concluded that after contractionary monetary policy shocks, oil prices and output falls maximized around the third quarter. Pagliacci (2019) revisited the Fry and Pagan (2011) model for seven Latin American countries and the United States. They used two variable structural models to diagnose the impacts of supply and demand disturbances in each country. One of the primary variables in this work was the output gap, extracted from real gross domestic product series through the HP filter method, which is suggested as a more reliable tool than differencing real GDP series. The main finding derived from this study is that the majority of the output gap and inflation variability in both the short and long terms is tied to supply-side shocks.

Although there is some research regarding monetary policy shocks for the Russia case, to the best of our knowledge, there is no article on the price puzzle problem for this country. Vdovichenko and Voronina (2006) investigated the effects of policy interest rates on the interbank market for Russia for the post-crisis period. The GMM results provided weak evidence for this relationship. Similarly, Korhonen and Nuutilainen (2017) utilized the GMM methodology to characterize Russian monetary management. They reported the validity of the Taylor Rule between 2006–2012. This finding is consistent with the inflation targeting policy of the Russian Central Bank and accepts the supreme role of the interest rates. One of the latest studies concerning the impacts of monetary policy shocks on the Russian economy was conducted by Nguyen et al. (2017). They examined the interactions between commercial lending rates and monetary policy rates and could not find any evidence for a long-run relationship between these variables. Using principal component analysis and VAR for BRIC countries, Cekin et al. (2019) found evidence that interest rate hikes create a negative impact on the stock market for the Russian economy. They stated that the higher yield on Chinese bonds diminishes demand for Russian bonds despite the increasing tendency for the equities. Kreptsev and Seleznev (2018) pointed out that under the fixed exchange rate regime, a one percent increase in interest rates leads to a half a percent increase in inflation within the four-forecast horizon. Under the floating exchange rate regimes, the increase in inflation is around one and a half percent. Puah et al. (2019) confirmed the short-run trade-off between inflation and output for Russia. Overall, these studies provide strong evidence regarding the critical role of monetary policy shocks

in the macroeconomic aggregates of the Russian economy. However, the authors employed conventional methods that have limitations. Our study aims to provide valuable insights into the literature by using a contemporaneous method based on theoretical foundations to investigate the effects of monetary policy shocks on Russian economic fundamentals.

## 3. Data and Methodology

### 3.1. Data

We collected all the data used in this research from the Main Economic Indicators Publication Database by the Organisation for Economic Co-operation and Development (OECD 2019). Data consist of quarterly series from Q1 of 1997 to Q1 of 2019 with 116 observations, and all series were transformed into logs. We derived average weighted 31–90 days interbank RUB loan interest rates to represent short-run interest rates. This variable has been used extensively in the recent literature (Freixas and Jorge 2008; Lee 2009; Panagopoulos and Tsouma 2019; Wang 2019), which emphasized the fundamental role of the interbank rates in output by generating a liquidity shock on the credit market. Then, we chose the consumer price index (CPI) as a proxy for inflation, which is indexed in (2015=100). It measures the general price movement by examining a weighted average of prices of some ordinary goods and services within a basket. This variable is the main measure of the cost of living and was also mentioned by Cho and Moreno (2006) as a robust indicator. Lastly, the real GDP represents the total value of all goods and services produced in one year indexed (2015=100) chosen by following suggestions of Farmer and Nicolò (2019).

### 3.2. The Concept of SVARs

This subsection aims to provide knowledge regarding one of the most popular instruments for assessing economic policy implications, structural vector autoregressive (SVAR) models. This model is based on a VAR model that has a useful tool called the impulse response that is used to interpret the response of the variables to an arbitrary shock (Sims 1980). However, a basic VAR model has no restrictions, which means it does not include any institutional knowledge (Yıldız et al. 2021). In this respect, SVAR is an improvement of the basic VAR model. Inoue and Kilian (2013) stated that SVAR is a reduced form of VAR model identified by researchers as representing prior knowledge of the system. Kilian (2011) emphasized that the main ambition of the studies based on SVARs is identification; they demand identification restrictions triggered by knowledge in the economic theory. Kilian and Lütkepohl (2017) highlighted the SVARs used to examine the patterns of the impulse responses of the model to infer policy implications.

To understand SVAR, it would be useful to start with a simple VARs equation demonstrated below:

$$z_t = A_1 z_{t-k} + e_t, \tag{1}$$

in which $A_1$ denotes n × k matrices of coefficients, $z_t$ is an n × 1 vector of variables, and $e_t$ is a reduced form of shock with zero mean expectation and constant covariance matrices without serial correlations. This leads to the SVAR representation of the variables:

$$B_0 z_t = B_1 z_{t-1} + \epsilon_t, \tag{2}$$

in which $\epsilon_t$ represents independent and identically distributed shocks, while there is no correlation between any of them. The objective is to obtain structural innovations $\epsilon_t$ by using both Equations (1) and (2) (Fry and Pagan 2011). The structural innovations are linear combinations of VAR errors. There is a parsimonious need for weights on the estimated B matrix for obtaining the structural innovations. Consequently, the model could have the following MA representation with the spread of structural shocks formed through impulse response functions:

$$z_t = D_0 e_t + D_1 e_{t-1} + \cdots + D_k e_{t-k}, \tag{3}$$

where $D_k$ represents the kth lag impulse response of $z_{t+k}$ to a unit change in $e_t$. Thus, the MA representation of SVAR is

$$z_t = C_0 e_t + C_1 e_{t-1} + \cdots + D_k z_{t-k} e_t,\tag{4}$$

where the kth period subjected innovations impulse responses of $z_{t+k}$ are calculated by $C_k = D_k B_0^{-1}$.

### 3.3. Sign Restricted SVARS for Open Economy

The new Keynesian foundation of our model is based on the works of Cho and Moreno (2002, 2006). Their contribution to economic literature by building structural equation modelling is to increase understanding of the dynamics of interest rates ($i_t$), inflation ($\pi_t$), and output gap ($y_t$). Therefore, the model we used is based on the equation systems given below (Ouliaris et al. 2016):

$$y_t = \alpha_{11} y_{t-k} + \beta_{12} i_{t-1} + \mathfrak{N}_{13} \pi_{t-1} + e_{1t},\tag{5}$$

$$\pi_t = \alpha_{21} y_{t-k} + \beta_{22} i_{t-1} + \mathfrak{N}_{23} \pi_{t-1} + e_{2t},\tag{6}$$

$$i_t = \alpha_{31} y_{t-k} + \beta_{32} i_{t-1} + \mathfrak{N}_{33} \pi_{t-1} + e_{3t}.\tag{7}$$

Two of the prominent shocks in Equations (5) and (6) are, respectively, a demand and cost–push shocks. The last equation provides shocks from monetary policy. The fundamental logic of the model is based on the assumptions that interest rates do not affect the output gap for the concurrent period; that there are no immediate impacts of current interest rates upon its effect with a lag; and last but not least that interest rates governed by monetary authority respond to the current inflation including output gap. Nonetheless, solely implementing restrictions is not adequate for avoiding the price puzzle. Therefore, there is a necessity for a different approach to avoid impulse response anomalies.

Our analysis is based on the conceptual framework of Fry and Pagan (2011), which focuses on the rotation matrices (SRR). In addition to it, Ouliaris and Pagan (2016) employed two alternative procedures together: sign restriction with generated coefficients (SRC) and rotation matrices (SRR). One advantage of the introduced approach is that it allows for applying sign restrictions on arrangements of equations that are partly identified. It is proper for merging zero restrictions with the sign restrictions. Moreover, it supports narrowing the span of impulse responses by normalization employed on median responses. Below, Table 1 illustrates the sign restriction relationships between variables and structural shocks.

**Table 1.** Sign Restrictions for Structural Shocks.

| Variable/Shocks | Demand | Cost–Push | Monetary Policy |
|:---:|:---:|:---:|:---:|
| $y_t$ | + | − | − |
| $\pi_t$ | + | + | − |
| $i_t$ | + | + | + |

Note: Entries show the imposed restrictions: + and − for positive and for negative signs respectively.

The SRR approach commences with IRs with uncorrelated shocks consisting of $cov(e_t)$ = $\Omega_R$ with unit variances. One might attain uncorrelated shocks $y_t = Pe_t$, by using CD (Cholesky decomposition) or SVD (singular value decomposition). According to CD, A is a triangular matrix, $\Omega_R = AA'$; thus, $(A')^{-1} = P$ provides uncorrelated shocks that can be converted to shocks with unit variance. Similarly, for SVD, the process proceeds as $\Omega_R = UFU'$, where $UU' = I$, $UU' = F$. Setting diagonal matrix $P = U'$ will produce uncorrelated shock by matrix F, and similarly, it can be expressed in terms of unit variances. After obtaining these variances, it is possible to obtain IRs of them, and this allows us to recombine the initial set. Furthermore, IRs might be attainable by a square matrix Q, featured $QQ' = I_n$ to provide uncorrelated innovations. There is more than one suggestion

for the derivation of Q matrices including the Givens matrices. Those who maintain IRs consort with sign restrictions. Another way is offered by Rubio-Ramirez et al. (2010), who use a simulation algorithm that has limited features. In the aforementioned practices, Q matrix applications cannot be practised with both short and long-term parametric SVAR restrictions. The SRR approach is based on recombining sign restriction initial information by reconsolidating Givens and Householder approaches to build a new set of orthogonal shocks. The primary standardized innovations are reproduced to develop a fresh set of innovations, from which an extra set of impulse responses is obtained and decided against the signs. The process of generating multiple responses is repeated many times until the system protects those that meet the sign constraints.

To exemplify the SRC method, we regress $\pi_t - \beta_{22}i_{t-1} - \mathfrak{N}_{23}\pi_{t-1} - e_{2t}$ on other variables using $e_{1t}$ as the instrument for $y_t$ to attain $e_{2t}$. The same procedure should be applied to Equation (7) using $e_{1t}$ and $e_{2t}$ on behalf of $y_t$ and $\pi_t$ to obtain $e_{3t}$. To create the widest potential variety of impulse responses, in respect of sign restrictions, the values of the coefficients generated as in the work of

$$\alpha_{11}\theta_1 = \left( \frac{\theta_1}{1 - abs\theta_1} \right), \ \beta_{12}\theta_2 = \left( \frac{\theta_2}{1 - abs\theta_2} \right) + \cdots +, \ \mathfrak{N}_{kk}\theta_k = \left( \frac{\theta_k}{1 - abs\theta_k} \right) \quad (8)$$

where $\theta_k$ represents a $(-1, 1)$ uniform probability density function and is an absolute value abbreviated by abs for each $k = 1, \ldots, 6$. After obtaining $\theta_k$'s, the estimated equations are later denoted with respect to all the variables; consequently, the calculation of the impulse response functions to one-standard-error shocks available for the applier. The process of the estimation will proceed to draw until the parameters satisfy the model. An additional advantage with this methodology is that it proposes obtaining median impulse responses, i.e., a summarization of the large aggregates drawn from impulse responses.

To highlight the median responses, Fisher and Huh (2019) demonstrated median responses that compiled an enormous number of agreed-upon IRs and included establishing the allowed responses at any forecast horizon in a soaring sequence and picking the 50th percentile responses. The issue that emerged from any percentile of median responses was that drawn from a specific SVAR coefficient. In each derivation from one forecasting horizon to the following, it was anticipated to evolve from various models with various $\theta_k$ parameters, which could lead to multiple models' problems. To cope with this problem, Fry and Pagan (2011) proposed a median target response solution. They suggested a criterion for obtaining a unique SVAR whose IRs were nearest to the median responses. Ouliaris and Pagan (2016) developed median target responses with incorporated IRs by maxima.

Although this empirical method can subdue the price puzzle problem, it has certain limitations while drawing the widest spectrum of presumable responses (Ouliaris and Pagan 2016). Specifically, for the SRR approach, it might happen by distinct twofold characteristics. The generation of median responses can prompt the selection process of a beginning set of IRs. Additionally, the Q matrix itself could lead to a limitation based on settlements of Givens and simulative procedures. It is possible to develop healthier essential settlements through future studies.

## 4. Empirical Analysis

### 4.1. Preliminary Tests

Before sharing the sign-restricted SVAR result, several tests were implemented to analyse the properties of series by descriptive statistics, autocorrelations by portmanteau test and LM test, and the information criteria to determine lag lengths. Following that, the presence of unit roots on series was investigated. Table 2 provides descriptive statistics of the subjected variables. According to Jarque Bera's test, all the subjected series are not normal. Table 3 provides the results of VAR lag length information criteria, portmanteau tests, and VAR residual serial correlation LM autocorrelations tests (Davies and Newbold 1979). It is evident from Table 3 that the best lag length is symbolized by the sign "*", which is the 3rd lag with a consensus of AIC ($-11.29$) and HQ ($-10.94$) information criteria.

Furthermore, the LM test results beyond three lags do not reject the null hypothesis of no serial correlation. That is supportive evidence that our lag selection is sufficient in terms of these two information criteria. A few caveats to the readers' attention are that the em dashes "—" under the column of portmanteau tests indicate that it is not able to give valid results up to three lags because it is the selected lag order for VAR.

**Table 2.** Descriptive Statistics of the subjected variables.

|  | INFL | INT | GAP |
|---|---|---|---|
| Mean | 3.78 | 2.28 | 0.01 |
| Median | 3.92 | 2.17 | 0.02 |
| Maximum | 4.74 | 3.53 | 0.07 |
| Minimum | 1.92 | 1.44 | −0.08 |
| Std. Dev. | 0.77 | 0.51 | 0.02 |
| Jarque-Bera | 10.5 | 8.85 | 7.25 |
| Probability | 0.00 | 0.01 | 0.02 |
| Observations | 87 | 87 | 87 |

**Table 3.** Lag Length Criterion and Autocorrelation Tests Results.

| Lag Length | AIC | SC | HQ | LM Test Result | Portmanteau Test Results |
|---|---|---|---|---|---|
| 1 | −10.20 | −9.93 | −10.14 | Serial Correlation | — |
| 2 | −11.04 | −10.44 * | −10.80 | Serial Correlation | — |
| 3 | −11.29 * | −10.42 | −10.94 * | No Serial Correlation | — |
| 4 | −11.225 | −10.10 | −10.77 | No Serial Correlation | No Serial Correlation |

(*) indicates the appropriate lag length.

What stands out in Table 4 is the findings of conducted unit root tests to examine the integration of the variables (Shin and Schmidt 1992). The tests without intercept specification might show series that contain unit roots due to misspecification. However, the tests with statistically significant intercept terms show that the variables are stationary in levels.

**Table 4.** Unit Root Test Results.

| Variables | Levels | | | | | First Differences | | | | |
|---|---|---|---|---|---|---|---|---|---|---|
|  | ADF None | ADF Int. | PP None | PP Int. | KPSS Int. | ADF None | ADF Int. | PP None | PP Int. | KPSS Int. |
| INFL | 1.43 | −5.99 * | 3.08 | −4.05 * | 1.12 | −1.99 *** | −3.06 * | −2.90 * | −3.78 * | 0.61 *** |
| INT | −1.31 | −2.67 *** | −1.31 | −2.70 *** | 0.47 *** | −8.26 * | −8.24 * | −8.26 * | −8.25 * | 0.13 |
| GAP | −4.37 * | −4.35 * | −3.54 * | −3.53 * | 0.03 | −5.67 * | −5.64 * | −5.73 * | −5.70 * | 0.06 |

Notes: $* \ p < -0.01$; $*** \ p < 0.01$ significance level.

### 4.2. Variance Decomposition and Median Responses

In this subsection, we estimate both the median target responses and median responses following Ouliaris and Pagan's (2016) specification. Additionally, the forecast variance decomposition (FEVD) is provided to differentiate the impact of each structural shock on the three macroeconomic aggregates. According to the FEVD presented in Table 5, demand shocks account for 79% of fluctuations in the output gap. While supply shocks contribute

10%, monetary policy shocks do not appear to be effective. Apparently, monetary policy shocks are ineffective in the fluctuations in the real sector.

**Table 5.** Forecast Variance Decomposition of Structural Shocks.

| Variable/Shock | Time | Demand | Supply | MP |
|---|---|---|---|---|
| GAP | 2 | 93 | 6 | 1 |
| | 5 | 82 | 15 | 3 |
| | 8 | 80 | 16 | 4 |
| | 10 | 79 | 18 | 3 |
| Inflation | 2 | 9 | 90 | 1 |
| | 5 | 25 | 71 | 4 |
| | 8 | 25 | 70 | 5 |
| | 10 | 22 | 71 | 6 |
| Interest | 2 | 10 | 3 | 87 |
| | 5 | 10 | 12 | 77 |
| | 8 | 19 | 20 | 61 |
| | 10 | 23 | 22 | 55 |

The impact of supply shocks on inflation is from dropped 20% to 71% but is still the most effective in the long run. Conversely, the impact of demand shock rises to 20% in the long run. The monetary policy shock is the dominant force (87%) in the 2nd forecast horizon in interest rates. However, its long-run impact declines to 55%. From these data, we can see that productivity shocks are dominant in fluctuations in real wages. The long-run impacts of demand and supply shocks on interest rates areis almost equal for Russia.

The median target responses acquired by the SRC approach are presented in Figure 1. Examining the median target responses, a demand shock has no permanent effects on output and is small compared with other disturbances. Following a demand shock, the response of inflation is positive, and there is not an instant fall. This impact is consistent with a shift in the IS curve. Moreover, the demand shock causes a significant rise in interest rates that dies out in the long run. What is striking from the median target due to the cost–push shock on the output gap for Russia is that it significantly decreases output at the beginning. This impact dies out around seven quarters. The effect of a cost–push shock on the price level was found to be permanent. The response of interest rates to cost–push shocks immediately peaked on the second forecast horizon and converged to zero in the long run. After a cost–push shock, the response of interest rates falls around four forecast horizons. Lastly, the response of output gap to monetary policy shock is just a trace amount peaked around the third and fourth forecast horizons. The clear benefit of our sign-restricted approach is overcoming the price puzzle. The monetary policy shock generates lower inflation of roughly 2.7% on the second forecast horizon. In addition, median target response functions show that after positive monetary policy shocks there is a negative impact on interest rates. In addition, median target response functions show that there is a negative impact on interest rates following monetary policy shocks.

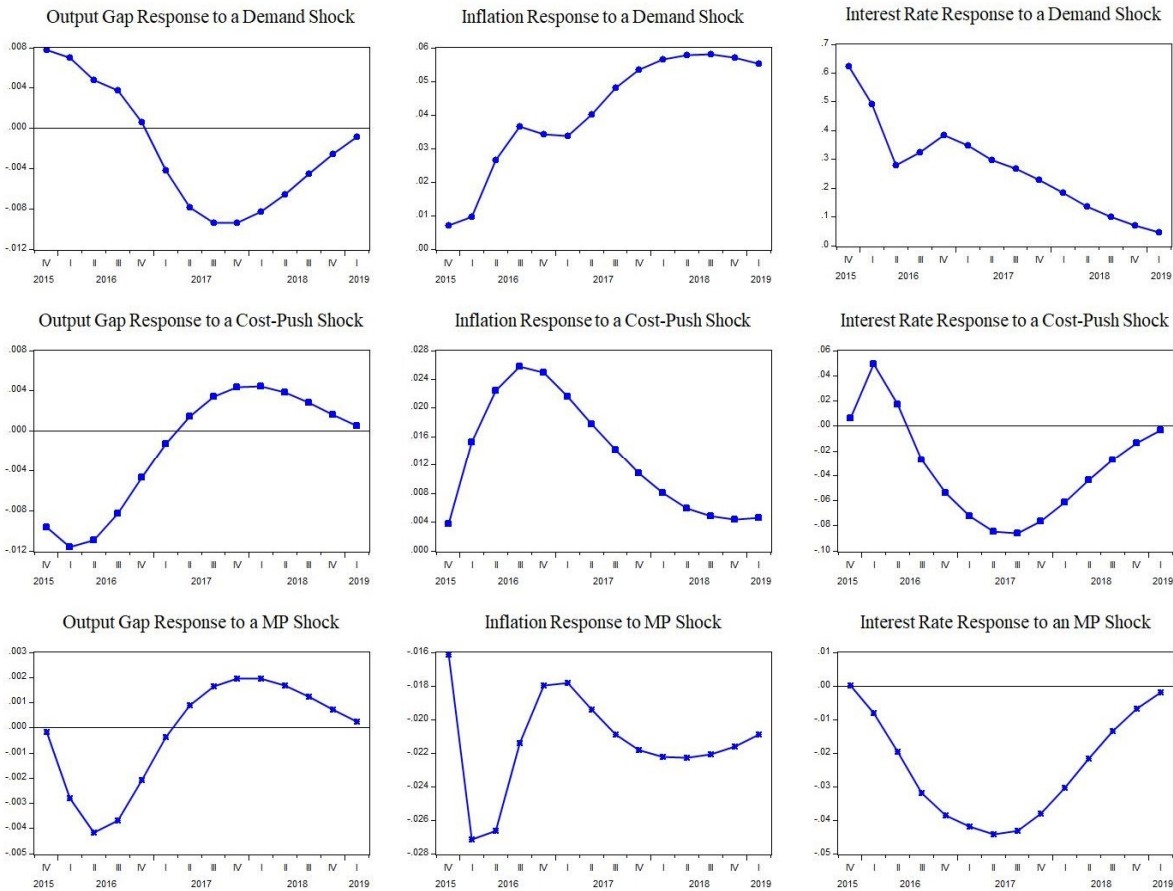

**Figure 1.** SRC Median Target Impulse Responses.

Figure 2 illustrates the median responses generated by the SRR approach for the Russian economy. The responses of the output gap to demand shocks are similar to those for the median target responses. A shock generates a peak at the beginning, but that small positive impact dies out in the fifth forecast horizon. The pass-through mechanism of a demand shock on inflation is persistent in a perplexing manner. Moreover, a demand shock generates a notable rise in interest rates, converging to zero on the long-time horizon. Cost–push shocks generate a significant drop in the economic performance of Russia. The detrimental effect of cost–push shock on output peaked between two and three forecast horizons but died out in the long run. That also generated more significant rises in inflation relative to demand shock in the Russian economy. Finally, the transmission mechanism of monetary policy shock in output is similar to median target responses. It generates limited drop downs on output that converge to zero around the fifth forecast horizon. Following monetary policy shocks, there is a significant drop in inflation. In contrast to median target responses, there is a short-run rise in interest rates after a monetary policy shock.

Overall, our results suggest that monetary policy shocks have long-run neutrality, which is supported by Gerlach and Smets (1995) and Fisher and Huh (2019). The fact is that supply-side shocks present a weaker form than demand-side shocks in inflation for Russia, due to its being an energy supplier of the world.

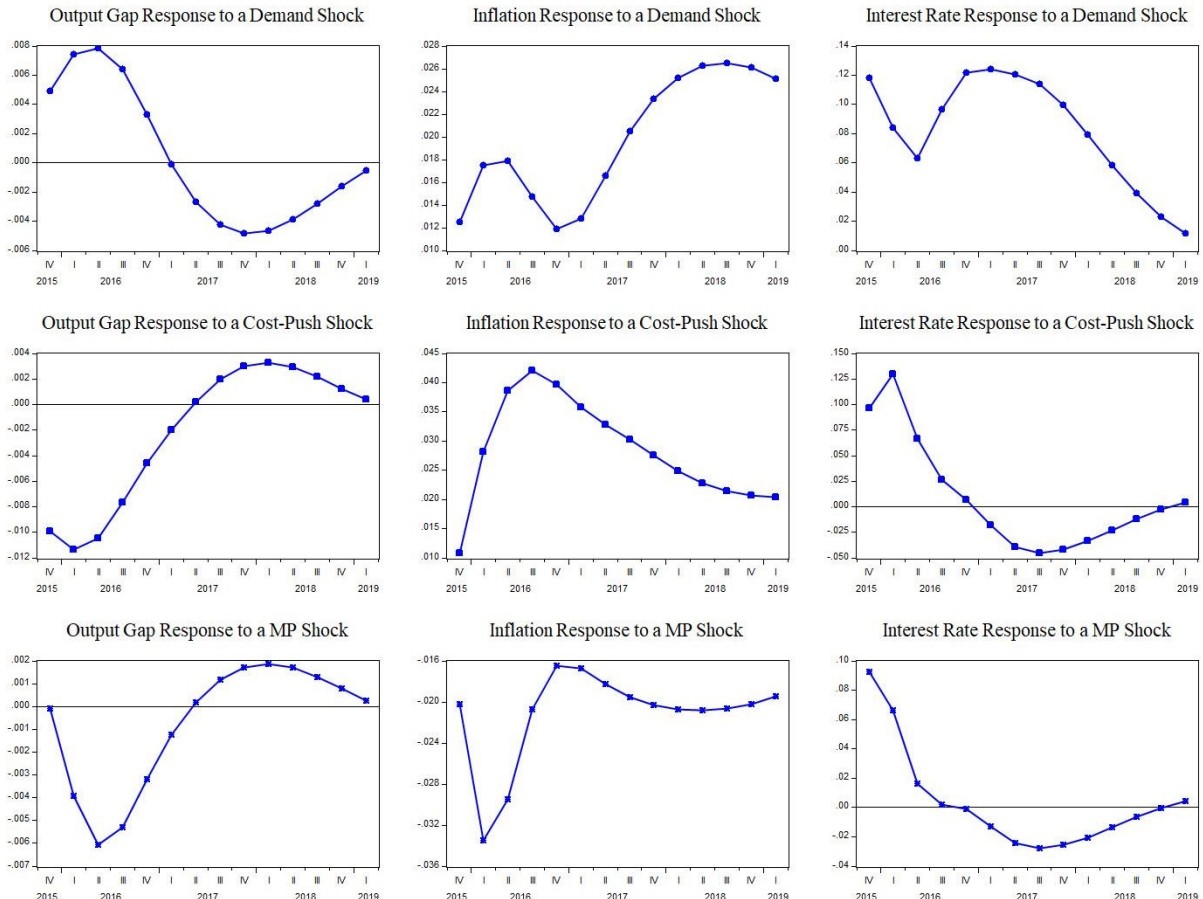

**Figure 2.** SRR Median Impulse Responses.

## 5. Conclusions

Even though many studies have examined the impacts of monetary policy on macroeconomic aggregates, most of the studies suffered from misleading impulse responses that caused unexpected findings such as price puzzles. Although the literature offered some solutions to this problem, none of them produced satisfactory results. To overcome the limitations in the existing literature and to fill the gap in the literature, in this paper, we used both parameter and sign restrictions for the first time to investigate the effects of policy shocks on the economic aggregates for Russia by implementing SVARs based on new Keynesian dynamics, including sign restrictions.

Our findings showed that although the effects of demand shocks on inflation are persistent, their effects on the output are limited and temporal. Additionally, cost–push shocks have significant damaging effects on both inflation and output. Interest rates respond strongly to both cost–push and demand shocks. On the other hand, monetary policy shocks have limited and temporal negative impacts on the output gap, although they cause a significant drop in inflation in the short run. Additionally, we considered the impacts of monetary policy shocks by using both median target and median responses and successfully solved the price puzzle problem. The findings of this study suggest that using both approaches together, demand shocks do not create any significant impact on the output gap, but the impacts of monetary policy shocks on inflation are more persistent than the output gap. A possible explanation for the results is that a tight monetary stance could generate a tolerable diminishing effect on the output gap that is robust in controlling inflation. Additionally, the evidence from this study justifies the new Keynesian theory that monetary policy shocks only have short-run effects.

Moreover, our results imply that Volcker–Greenspan's rule (Clarida et al. 2000) could be a useful guide for policy makers to solve the problem efficiently. The rule emphasizes

the importance of re-establishing nominal expectational stability to ensure that the price system works properly and removes the cyclical inertia in the interest rate (Hetzel 2007). In addition, Clarida et al. (2000) suggest that the Volcker–Greenspan baseline policy reaction function should be estimated using all the available information, including the expectation of the future values of inflation and output gap. Given these factors, policy makers convince economic actors that they will use the interest rates when necessary. Although the Russian Central Bank applies the optimal policy, as a result of the political tensions facing Russia, it needs more legal and financial support than other institutions. All governmental institutions should provide the necessary support for inflation targeting, which was being carried out by Russian authorities in 2017 to convince the market. Since the measures taken by the Russian government were very cautious, the provision of precautionary measures reduces growth and loses competitiveness in world trade. A reasonable approach to this problem should be an emerging concern of Russia's. Taken together, the median responses and median target responses provide results that are in the same direction. The only difference is that median responses are slightly wider compared with median target responses owing to the impulse response that draws the generation process. Interestingly, the only difference was in the response of interest rates to monetary policy shocks, which can be seen in different studies for example by Fisher and Huh (2016, 2019). In terms of both responses, this study provides a better solution to the price puzzle problem.

In terms of future work, researchers could include fossil fuel prices to shed more light on understanding the dynamic interactions of monetary policy and main macroeconomic aggregates in the Russian case and other countries. Since fossil fuels constitute the majority of foreign trade revenues in Russia, the evaluation of structural shocks in oil prices will make a significant contribution to the empirical literature. Furthermore, the analysis of the effect of the exchange rate transmission mechanism on the prices could also be beneficial. The approach used in this paper is fresh in the literature; therefore, extensions of our study could include applying the approach used in our paper to study whether it works for other countries. This study applies both the new Keynesian Model and a sign- and parametric-restricted model to investigate the effects of policy shocks on the economic aggregates for Russia by implementing SVARs. Academics could use more advanced models to extend our work. Extensions of our study could also include applying our approach to studying other important issues, for example, anomaly (Guo et al. 2017), profitability (Nguyen et al. 2020), stock returns, and volatility (Zada et al. 2021). Readers may refer to, for example, Wong (2020) and Woo et al. (2020) to know more about other important issues that our approach could be used to study.

**Author Contributions:** Conceptualization, methodology, validation, writing—review and editing, supervision, project administration K.K.G. and W.-K.W. Software, formal analysis, investigation, data curation, writing—original draft preparation, visualization B.F.Y. All authors have read and agreed to the published version of the manuscript.

**Funding:** This research received no external funding.

**Institutional Review Board Statement:** Not applicable.

**Informed Consent Statement:** Not applicable.

**Acknowledgments:** The authors thank Ralf Fendel, and the anonymous referees for their helpful comments which help to improve our manuscript significantly. The third author would like to thank Robert B. Miller and Howard E. Thompson for their continuous guidance and encouragement.

**Conflicts of Interest:** The authors declare no conflict of interest.

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
