# Peer review of "Analysing Monetary Policy Shocks by Sign and Parametric Restrictions: The Evidence from Russia"

_economies, doi:10.3390/economies10100239_

Round 1

Reviewer 1 Report

In my opinion it is a very well-written and thoughtful article. The study made an appropriate selection of tools for its conduct and econometric tests.

A wide range of literature was used - significant authors on this topic.

One can only fault the abstract that it lacks a clearly defined research objective and the discussion could be developed a little more. It is worth highlighting the limitations of this study.

Author Response

Thank you very much for your invaluable comments and suggestions, which have improved the revised version significantly.

We would also like to send our appreciation to you for your time and efforts in reviewing our paper. We would like to thank you for your following comments:

  • Does the introduction provide sufficient background and include all relevant references? (yes)
  • Are all the cited references relevant to the research? (yes)
  • Is the research design appropriate? (yes)
  • Are the methods adequately described? (yes)
  • Are the results clearly presented? (yes)
  • Are the conclusions supported by the results? (yes)
  • English language and style are fine/minor spell check required
  • I don't feel qualified to judge about the English language and style
  • In my opinion it is a very well-written and thoughtful article.
  • The study made an appropriate selection of tools for its conduct and econometric tests.
  • A wide range of literature was used - significant authors on this topic.

We would also like to send our appreciation to you for your time and efforts in reviewing our paper and for providing excellent comments. Below are our responses to your helpful comments and suggestions.

Question 1. One can only fault the abstract that it lacks a clearly defined research objective and the discussion could be developed a little more.

Answer 1:  Thank you very much for your advice. We have stated clearly in the abstract the defined research objective and developed the discussion further in our revised manuscript.

Question 2. It is worth highlighting the limitations of this study.

Answer 2:  Thank you very much for your advice. We have highlighted the limitations of this study in our revised manuscript.

We hope that you will find this manuscript suitable to be included in an upcoming issue of your publication.

Reviewer 2 Report

See attached PDF file.

Author Response

Thank you very much for your invaluable comments and suggestions, which have improved the revised version significantly.

We would also like to send our appreciation to you for your time and efforts in reviewing our paper. We would like to thank you for your following comments:

  • Does the introduction provide sufficient background and include all relevant references? (yes)
  • Are all the cited references relevant to the research? (yes)
  • Is the research design appropriate? (yes)
  • Are the methods adequately described? (yes)
  • Are the conclusions supported by the results? (yes)
  • In my opinion it is a very well-written and thoughtful article.
  • The study made an appropriate selection of tools for its conduct and econometric tests.
  • A wide range of literature was used - significant authors on this topic.
  • The paper is well researched and well motivated.
  • The authors cite plenty of relevant literature, including Cho and Moreno, who derive the steps required to go from an NK model

We would also like to send our appreciation to you for your time and efforts in reviewing our paper and for providing excellent comments. Below are our responses to your helpful comments and suggestions.

Question 1. Are the results clearly presented?

Answer 1:  Thank you very much for your advice. We have presented the results clearly in our revised manuscript.

Question 2. Extensive editing of English language and style required

Answer 2:  Thank you very much for your advice. We have polished our paper carefully in our revised manuscript.

Question 3. Based on previous work by S. Ouliaris, A.R. Pagan, and J. Restrepo, the authors estimate the reduced form of a three-equation structural vector auto-regressive (SVAR) model of the Russian

economy to simulate via impulse responses the effects on inflation, output, and interest rates of

selected supply, demand, and monetary shocks in order to address what is known in the literature

as the ”price puzzle,” the seeming rise in prices immediately following an unexpected tightening

of monetary policy

Answer 3:  Thank you very much for your advice and information. We have addressed the issue as highlighted in blue in in our revised manuscript.

Question 4. Using a combination of parameter and sign restrictions to derive their SVAR, the authors seem to have hit on an empirical model that has nearly vanquished the elusive price puzzle, finding that while monetary policy shocks are non-neutral in the short run, as one would expect from the NK model, they are neutral to the real sector in the long run. And, importantly, such shocks are followed by reduced inflation in the short as well as in the longer run. In addition, they find that demand shocks do not significantly impact the output gap, while cost shocks have detrimental effects on output and inflation.

Answer 4:  Thank you very much for your advice. We have addressed the issue in our manuscript.

Question 5. I would urge the authors to delve more thoroughly into a discussion of the meaning of the unit-root tests in Table 4.

Answer 5:  Thank you very much for your advice. We have addressed the issue as highlighted in blue in in our revised manuscript.

Question 6. It is difficult to judge the quality of the estimated model, the various p-statistics not withstanding. The sample may be too small, but the empirical quality of a model is best judged by its ability to forecast, where Theil’s U-statistic applied to out-of sample predictions would be an appropriate metric. Is that a possibility? The reader will have to trust that the empirical results have some enduring value.

Answer 6:  Thank you very much for your advice. We have 116 observations in our study. Thus, our sample is large enough to draw meaningful conclusions.

Question 7. Finally, it is obvious from reading this fine paper, that the writers are non-English speaking. The entire text is is serious need of editing to bring it up to the standards of Economies.

Answer 7:  Thank you very much for your advice. We have polished our paper carefully.

We hope that you will find this manuscript suitable to be included in an upcoming issue of your publication.
